# Psychomotor and visual skills underlying position specialization in 1352 elite youth baseball players

**Jessica Ho[1], Sicong Liu[2], Zeyu Feng[3], Lawrence Gregory Appelbaum [3] ***

**1** Department of Psychiatry and Behavioral Sciences, Duke University, Durham, North Carolina, United States of America, **2** Annenberg School of Communication, University of Pennsylvania, Philadelphia, Pennsylvania, United States of America, **3** Department of Psychiatry, University of California San Diego, La Jolla, California, United States of America

* greg@health.ucsd.edu

## Abstract

Baseball is an international sport with participation from tens of thousands of people world-wide. In the United States, the Prospect Development Pipeline (PDP) is a collaborative effort between Major League Baseball and USA Baseball to establish a developmental pipeline leading to the professional draft. Players participating in the PDP undergo comprehensive evaluations that measure athletic performance, speed-of-processing, visual function, and on-field talent. The present study evaluated data from 1352 elite junior male PDP participants (aged 14 to 21) who signed informed consent, collected between 2017 and 2020, to identify latent abilities and their association with player specialization. Data were first subjected to Exploratory Factor Analysis (EFA) to reduce the 22 measured variables to a smaller set of latent abilities. The resulting factors were evaluated using multiple linear regression to predict each factor using age, height, weight, and position. EFA revealed a combination of physical and psychomotor skills accounting for 52% of the overall variance that grouped into four abilities: grip strength, functional vision, explosiveness, and rapid decision-making. Regression analyses demonstrated that these skills are associated with position assignments, controlling for age, weight, and height, and revealed that outfielders are the most explosive, infielders perform best on psychomotor measures, and catchers perform best on functional vision tests (ps < 0.001). These findings indicate skills that contribute to player specialization, providing new information about the developmental trajectory of junior elite baseball athletes that can be used for scouting and player development.

## Introduction

For more than 150 years, baseball has captivated millions of people around the world through the challenge of hitting a pitched ball with a 2.6-inch diameter bat. The physical and mental demands of the sport are so great that a player who can get on base one out of every three plate appearances is considered among the best-of-the-best on offense. However, hitting a baseball is only part of the story. On defense players are separated into nine positions that can be

Miami Blvd, Suite 119 Durham, NC 27703
russellhartford@usabaseball.com.

**Funding:** This research was funded by grant support to L.G.A. through the United States Army Research Office [W911NF-15-1-0390]. https://www.arl.army.mil/who-we-are/aro/. The authors would like to state that the funders had no role in study design, data collection and analysis, decision to publish, or preparation of the manuscript.

**Competing interests:** The authors would like to state that they have no competing interests.

grouped into four broader categories: pitchers, catchers, infielders, and outfielders. These positions are grouped in such a way because each has unique demands with corresponding physical and athletic attributes [1, 2]. Pitchers pitch balls from the mound and seek to prevent batters from hitting the ball and reaching base. Catchers suggest pitch selections, receive all pitches from the pitcher, frame pitches close to the strike zone, and throw across the field to put out any base runners attempting to steal a base. Infielders are tasked with fielding balls hit off the bat at up to 120 miles per hour, then making quick decisions on where to throw across the field to get batters out before they reach base. Outfielders are responsible for covering an expansive field for hit balls and returning the ball to the infield to prevent runners from advancing bases [2, 3].

Baseball experts observe that each position requires a set of skills that distinguish and overlap with other positions [3, 4]. Physical and athletic traits indicative of potential elite pitcher performance include tall and mid-weight individuals with a strong lower bodies and extreme arm strength. Elite catchers tend to be heavier and demonstrate greater arm strength and heightened decision making, with less emphasis placed on height. Common characteristics across the infielders include players with high agility, quick responsiveness, and high-power output from the lower and upper extremities. Outfielders are tall, strong, powerful, and can quickly cover a lot of ground. Collectively, these views stem from the demands of position and many years of domain knowledge, however experts recognize the need for greater data driven insights in talent identification [5].

From the development of youth athletes to the most accomplished professionals, there is an unprecedented effort to cultivate skills that promote success. As a result, various physical [6], cardiovascular [7], psychological [8], and visual [9–12] assessments have emerged with the goal of identifying key measures that predict future success. Together, studies utilizing these measures have indicated that psychomotor and visual skills may contribute to on-field success, opening the door for such evaluations to be used in scouting and player selection. Further, interest in understanding the developmental trajectory of junior athletes has evolved substantially in the realm of sports science [13–16] and particularly in baseball with the emergence of "Sabermetrics", an empirical analysis of baseball statistics to evaluate player performance and develop playing strategy [17, 18].

The Prospect Development Pipeline (PDP) was established as the official talent identification pathway for elite amateur baseball players to all 30 Major League Baseball (MLB) clubs. As a joint venture between MLB and USA Baseball (USAB) this invites elite amateur athletes, based on past game performance, to participate in PDP sanctioned camps. At camps, athletes undergo the PDP Performance Assessment, a series of comprehensive evaluations developed from expert baseball professionals that use state-of-the-art technology to measure movement, cognitive skills, speed-of-processing, athletic performance, eye movements, and on-field talent. The results of these assessments are provided to each participant and are retained by USAB and MLB, allowing for a comprehensive set of metrics on the player's strengths and weaknesses, while also providing information about areas of potential improvement for future development. Participation in these camps also offers extensive visibility of players to MLB scouts, instruction from coaches, and educational programming for potential futures as professional athletes.

Identifying what talent selection criteria are best suited for a given position is not a new concept [19–21], but new sources of data and statistical analyses will contribute to the growth of this field. Given an available array of assessments, such as those carried out through PDP, discerning the groupings of skills helps players and coaches to identify strengths and weakness in one's skillset, which can help tailor training programs for optimal benefits. Further, showing the different skill profiles among players can help understand the development of specialized

expertise, such as those needed to excel at a given position. Yet this remains an under-investigated area in sports science with relatively few large-scale empirical studies to guide the process. Thus, the present study strives to fill this gap by testing the hypothesis that movement, strength, cognitive, and oculomotor skills differentiate players from different positions above and beyond information about age and physical attributes of the athletes. As such, this study helps understand player specialization to provide novel information about the developmental trajectory of junior elite baseball athletes.

In this article, we first introduce the dataset and procedures of each assessment performed during the PDP events. Next, we present the methods of analysis used to group the large number of measured tasks into a smaller set of latent skills and the approach to multiple linear regression for exploring correlations between latent skills and player positions. Results are then presented detailing findings followed by discussion of the implications and potential limitations of the findings.

## Materials and methods

The current study utilized data from 1352 junior elite male baseball players collected at PDP events from 2017 to 2020 as part of the USA Baseball, Prospect Development Pipeline. All individuals signed written research informed consent statements or if they were minors, consent was signed by their parent or guardian. Subsequent data was shared with the research team as part of a didactic data science collaboration. De-identified assessment scores were shared along with information about the event and date of participation and baseball-specific information for each individual. Ethnicity, social economic status, comorbidities, and sexual orientation were not collected. All data were shared with the research team, without inclusion of any protected health information, under a secondary-data protocol approved by the University Institutional Review Board that conformed to the tenants of the Declaration of Helsinki.

### Prospect development pipeline protocol and dataset

The PDP Dataset consists of multiple separate assessments measured in three test batteries as well as baseball participation information and player height and weight. All assessments are standardized and administered by a trained staff with consistent procedures across all venues. The dataset under consideration here was collected at 53 events occurring at 49 locations in the United States and Dominican Republic with players who were invited based on their past baseball performance and eligibility for the MLB draft.

The PDP Protocol is performed following initial registration and check-in at each event on a group of 20 to 25 players at a time through a circuit of brief stations set up on the field and in the field house that is completed in under one hour for each of the three batteries with breaks between the individual batteries. Each player was first measured on height and weight prior to completing the **Hand Grip Strength** battery, **Functional Eye Movement** battery, and **Kinematic** battery, in that sequence. Hand Grip Strength was assessed with a digital Jamar Hand Dynamometer™ in a standing position. The battery of kinematic assessments was collected using Microgate Witty SEMs™ sensors for the 30-Yard Sprint Assessment and Man-in-the-Box assessment, while the Microgate OptoJump Next™ was used to capture the Dynamic Broad Jump and Counter Movement Jump assessments. The battery of Functional Eye Movement assessments was performed with the RightEye™ eye tracking system on a computer with screen mounted eye tracking, against a white wall in a quiet room with no reflective surfaces. In the event that the athlete did not follow instructions, or the data was not accurately recorded, they were given an opportunity to repeat the assessment. Descriptions for each assessment and resulting data variables are provided below.

**Hand grip strength battery.** During the Hand Grip Strength battery athletes are instructed to compress a digital dynamometer to measure their maximum grip strength for each arm in pounds, in a set order, while standing with their arm in one of three positions. In the "Down" position the elbow is straight and arm is extended against the player's body towards the ground. In the "90" position, the upper arm is against the player's body and the elbow is flexed at a 90-degree angle parallel to the ground and the wrist is unrotated with palm pointing towards the midline of the body. In the "Up" position the arm is fully extended and pointed up, straight over the player's head. In sequence, grip strength is measured for **Left Down, Right Down, Left 90, Right 90, Left Up,** and **Right Up** without breaks between this sequence.

**Functional eye movement battery.** The Functional Eye Movement battery includes eight tests from the RightEye™ Dynamic Vision module and 11 tests from the Sports Vision EyeQ module. During these tests, the participants were seated in a stationary chair and asked to look at a 3D Vision monitor that was adjusted in height to be centered in front of their eyes and fitted with a 120 Hz eye tracker, keyboard, and wireless mouse for the participant to register responses. The screen luminance was set to $85 \mathrm{cd}/m^2$ and participant's heads were unconstrained during the test. The individual tests comprising the Brain and Sports Vision modules measured the amplitude, velocity, and precision of eye movements during tasks that require saccadic and smooth pursuit eye movements, as well as oculomotor reaction times to central and peripheral stimuli. Participants were instructed to track the dot on the screen or saccade to new dots and task scores were combined into proprietary summary scores that consisted of linear combinations of individual variables with weights ranging from 0.1 to 13% across metrics [22, 23]. Brain and Sports Vision modules measure the following summary scores;

○ **Brain Fixation Score** is a measure of the ability to keep the eyes still.

○ **Brain Pursuits Score** is a measure of the ability to precisely pursue moving objects with the eyes.

○ **Brain Saccades Score** is a measure of the ability to perform precise saccadic eye movements.

○ **Sports Functional Score** is a measure of static and dynamic acuity and contrast sensitivity.

○ **Sports Mechanics Score** is measure of oculomotor mechanics including fixations, saccades, and smooth pursuit.

○ **Sports Mind Eye Score** is a measure of processing and reaction time in choice and discriminant decision tasks.

○ **Sports On-Field Score** is a measure of on- and off-task times.

**Kinematic battery.** The kinematic battery consists of four separate assessments, each with a subset of component measures.

• In the 30-Yard Sprint assessment, athletes watch the Witty SEM™ LED count down from five to one, followed by a random pause of zero to five seconds, after which the LEDs turn green indicating the go signal for the athlete to sprint the length of a straight 30-yard track. Sensors on the track capture step-by-step gait and provide separate measures for the athletes' speed generation and acceleration. Measures under consideration for this report are.

○ **Reaction-to-Go** measures in milliseconds the time from the start indicator turning green to the moment the individual crosses the first gate at the start of the track.

○ **10-Yard Split** measures in milliseconds the time from the start indicator turning green to when the runner crosses the 10-yard line.

○ **30-Yard Total** measures in milliseconds the time from the start indicator turning green to when the runner crosses the 30-yard finish line.

• During the Counter Movement Jump (CMJ) assessment, athletes perform three consecutive two-legged vertical jumps in which they are instructed to use both legs to complete three vertical jumps as high and as fast as possible. This examines the repeatability of power output, dynamic stability, control, and explosiveness on the y-plane.

○ **CMJ Height** is the vertical jump height, in inches, averaged over three consecutive jumps.

○ **CMJ Ground Contact Time** is the time spent contacting the ground, in milliseconds, between vertical jumps during the CMJ assessment.

• In the Dynamic Broad Jump (DBJ) assessment the athletes are instructed to line up outside the OptoJump system, then take a small "bunny hop" into the system to begin the DBJ. The athlete is also instructed to complete a max-effort jump in the x-plane while generating power in the shortest amount of time possible from the "bunny hop". This captures an athlete's ability to produce power as quickly as possible to jump as far as they can using both legs.

○ **Broad Jump Distance** measures the athlete's initial landing spot from their bunny hop to their final landing spot in inches.

○ **Broad Jump Ground Contact Time** is the time spent contacting the ground, in milliseconds, between broad jumps during the DBJ assessment.

• During the Man-in-the-Box assessment, athletes are positioned in the center of four Witty SEMs 7 by 5 LED-equipped proximity sensors, situated approximately 3 feet off the ground on tripods and placed in the corners of a 10' by 10' space. Athletes perform two tasks in this configuration.

○ During the **Green Box** task, one of the four LED proximity sensors unpredictably displays a square green light until it is activated by the athlete moving their hand within 4 inches of the sensor, after which the green square immediately appears randomly on any one of the four sensors. This task measures the time it takes to complete 10 touches.

○ During the **Green 3** task a variety of symbols (arrows, numbers, letters) in red, green, or blue appear on one sensor at a time. The athlete is instructed to touch the sensors showing a green number "3" but not the other symbols or colors. This task measures the time it takes to complete 10 touches.

## Sample selection

An initial sample of 1903 male players was shared with the authors for preliminary evaluation. After restricting the age range to 14 to 21 years old and limiting consideration to the positions of catchers, infielders, outfielders, and pitchers for more consistent sample representation, the final sample included 1352 individuals. As such, players younger than 14, older than 21, designated hitters, utility players, and individuals with missing ages or position designations were excluded from further analyses. Missing data was present in 4.9% of the overall cells under consideration and was replaced with values calculated through Multiple Imputation with Chained Equations (MICE) in the R software package. Further consideration of the assessment variables was limited to non-redundant measures that adhered to normal or near-normal distributions, resulting in a final sample of 22 assessment variables that were further analyzed for subsequent interpretable effects. Among the final sample of 1352 individuals, position coding

was aggregated into four position groups based on their primary position coding with 255 catchers, 496 infielders, 331 outfielders, and 294 pitchers.

## Transparency and openness

Sample size, data exclusions, manipulations, and all measures in the study are discussed and justified in the following sections. All data, analysis code, and research materials are available upon request for at least the next five years.

This study constituted a real-world, "close-to-population" sample of over a thousand athletes to explore data-driven patterns (i.e., latent skills and their regressions on demographic and position factors), and therefore a priori power analyses were not employed.

Data were analyzed using R, version 4.0.3 (R Core Team, 2021), the package ggplot, version 3.2.1 [24], and the package psych, version 2.1.9 [25] as well as the SPSS statistical software. This study's design and its analysis were not pre-registered.

## Statistical analysis

Analyses consisted of two stages that involved applying Exploratory Factor Analysis (EFA) and multiple linear regression, respectively. Additionally, repeated measures analysis of variance (rANOVA) was performed, with Greenhouse-Geisser correction for sphericity, to evaluate the roles of hand position, side, and handedness preferences on grip strength scores after excluding 38 players who reported being ambidextrous.

**Exploratory factor analysis.** EFA was performed to reduce the 22 measured task variables to latent skills. Bartlett test of homogeneity of variances and the Kaiser-Meyer-Olkin (KMO) measure of sampling adequacy (MSA) were first performed to evaluate whether the data were appropriate for EFA. The Barlett test results indicated that the data were suitable for EFA. Consistently, the KMO procedure demonstrated an overall MSA estimate of 0.8, suggesting an appropriate sample size for EFA. Overall, running the EFA was iterative with factor extraction specified as "factor analysis" with minimum residual (minres). The factor rotation method of promax oblique (with Kappa = 4) was used, given that Promax is ideal for identifying the presence of a "simple structure" with large sample size that accounts for correlations between variables [26]. The eventual selection of factor number was based on jointly considering Kaiser's rule, parallel analysis, and conceptual interpretation. All the extracted factor scores were standardized so that positive scores indicate better skill.

**Multiple linear regression.** To explain the variability in each skill factor identified from the preceding EFA stage, a multiple regression model was fit to predict each skill factor using age, height, weight, and position. Age, height, and weight were centered and modeled as continuous predictors, whereas position was an indicator predictor with pitchers serving as the reference category. Therefore, the intercept ($b_0$) term in the following generic regression model represented the corresponding skill level of a pitcher who has an average age, height, and weight among baseball players in the current sample. Having such a concrete reference helped interpret other model coefficients in relative sense.

$Model_i$ :

$$FactorScore_i \sim b_0 + b_1 Age + b_2 Height + b_3 Weight + b_4 CPosition + b_5 IFPosition + b_6 OFPosition$$

## Results

### Sample characteristics and assessment scores

As illustrated in Fig 1A, the sample of 1352 individuals were comprised of 14- to 22-year-old athletes with a mean age of 17.1 years and a standard deviation of 1.7 years. The sample had

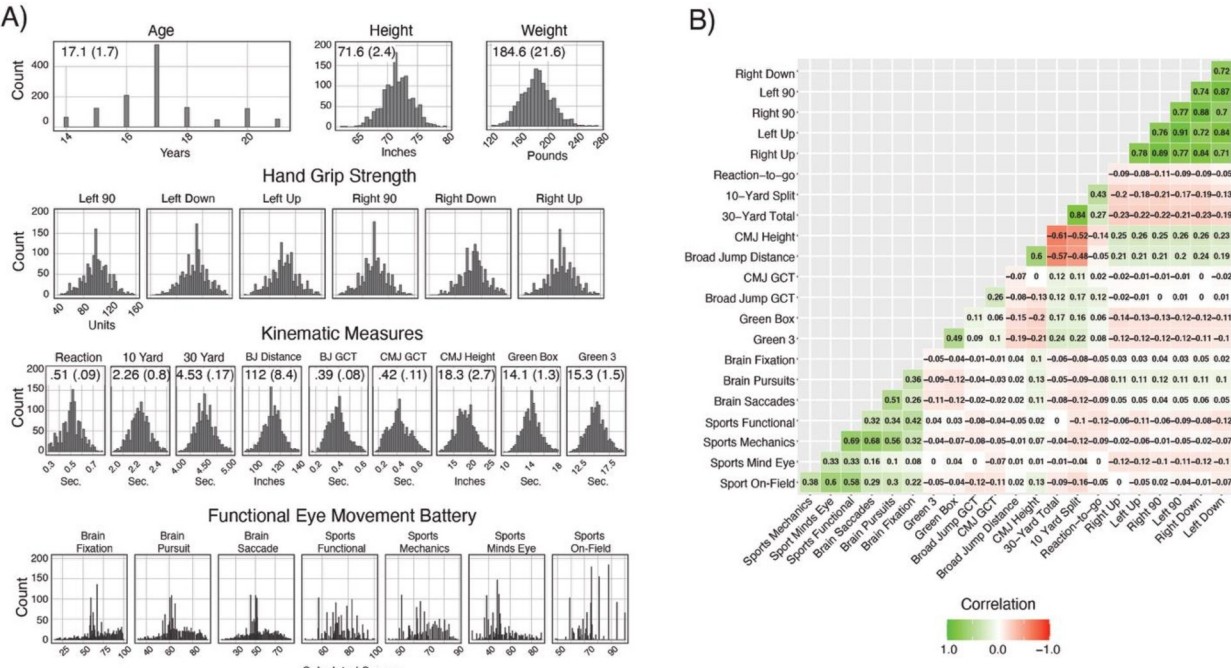

**Fig 1. A)** Histograms of age, height, weight, and the assessment variables, groups by task battery. **B)** Bivariable correlation matrix of assessment variables.

relatively more 17-year-old individuals. The average height was 71.6 inches, and the average weight was 184.6 pounds, with standard deviations of 2.4 and 21.6, respectively. While no individual metric was normally distributed (Kolmogorov-Smirnov p-values were all below 0.01), all scores all held relatively low skewness and kurtosis (Hand Grip Strength and Kinematic Measures between +/-.4 and Functional Eye Movement Battery between +/-0.8) and distributions approximated normal when accounting for the oversampling of 17-year-old players. As illustrated in Fig 1B, bivariate correlations between variables clustered with high coefficients between measurements collected among the same modality as well as some high correlations between modalities.

## Exploratory factor analysis

The EFA results supported a 4-factor model. **Table 1** displays the item loadings for the four factors with a threshold cutoff of 0.3. As illustrated in the table, the four factor EFA accounted for nearly 52% of the variance across 22 assessment variables with groupings that were interpretable given the specific composition of tasks that load onto each factor. The first factor accounted for 22% of the overall variance, was composed of the six, grip strength sub-scores, and was referred to as "Grip Strength". Loadings for this factor were all positive indicating that higher factor scores for an individual corresponded to greater grip strength. The second factor accounted for 13% of the variance, was composed of the seven functional vision sub-scores and was labeled as the "Functional Vision" factor. Loadings for this factor were positive indicating that higher factor scores for an individual corresponded to better performance on these tasks. The third factor accounted for 12% of the variance, was composed of Reaction-To-Go, 10-Yard Split, 30-Yard Split, CMJ Height, and Broad Jump Distance, and was referred to as "Explosiveness". Loadings on this factor were mixed with higher loadings associated with higher heights and greater distances, but lower times in the sprint measures. As such, higher

**Table 1. Factor loadings and eigen values of each assessment variable and explained variance for each promax rotated factor solution.**

| Assessment | Grip Strength | Functional Vision | Explosiveness | Rapid Decision-Making |
|---|---|---|---|---|
| Left 90 | 0.91 | | | |
| Left Up | 0.897 | | | |
| Right 90 | 0.893 | | | |
| Right Up | 0.888 | | | |
| Right Down | 0.867 | | | |
| Left Down | 0.857 | | | |
| Sports Mechanics Score | | 0.972 | | |
| Sports Functional Score | | 0.772 | | |
| Sports On-Field Score | | 0.641 | | |
| Brain Saccades Score | | 0.609 | | |
| Brain Pursuits Score | | 0.592 | | |
| Brain Fixation Score | | 0.422 | | |
| Sport Mind Eye Score | | 0.364 | | |
| 30-Yard Total | | | -0.98 | |
| 10-Yard Split | | | -0.929 | |
| CMJ Height | | | 0.641 | |
| Broad Jump Distance | | | 0.626 | |
| Reaction-to-go | | | -0.316 | |
| Broad Jump GCT | | | | |
| CMJ GCT | | | | |
| Green Box | | | | -0.739 |
| Green 3 | | | | -0.669 |
| Property | Grip Strength | Functional Vision | Explosiveness | Rapid Decision-Making |
| Sum of Squares loadings | 4.833 | 2.920 | 2.671 | 1.005 |
| Proportion Variance | 0.220 | 0.133 | 0.121 | 0.046 |
| Cumulative Variance | 0.220 | 0.352 | 0.474 | 0.519 |
| Proportion Explained | 0.423 | 0.255 | 0.234 | 0.088 |
| Cumulative Proportion | 0.423 | 0.678 | 0.912 | 1.000 |

Explosiveness factor loadings were interpreted as corresponding to better performance. The fourth factor, composed of the Green Box and Green 3 tasks, accounted for 4.6% of the overall variance and was interpreted as reflecting "Rapid Decision-Making" because these tasks required quick decision making and physical responsiveness. Loadings on this task were negative, indicating that higher factors scores are associated with quicker completion times.

## Multiple regression

Multiple linear regression was performed to evaluate the contribution of age, height, weight, and position (relative to pitchers) on the four factor loading scores. Each model was significant and as illustrated in **Fig 2**, many individual variables were significant within each model. Descriptions of each model results are as follows.

**Grip strength.** This model produced a significant ($p < 0.001$) fit with the independent variables together accounting for 20% of the variance in Grip Strength factor loading scores. Increases in age, height, and weight were each significantly associated with higher grip strength. Catchers, infielders, and outfielders each showed significantly higher grip strength compared to pitchers, while holding all other variables constant. Outfielders had the highest average grip strength when compared to pitchers.

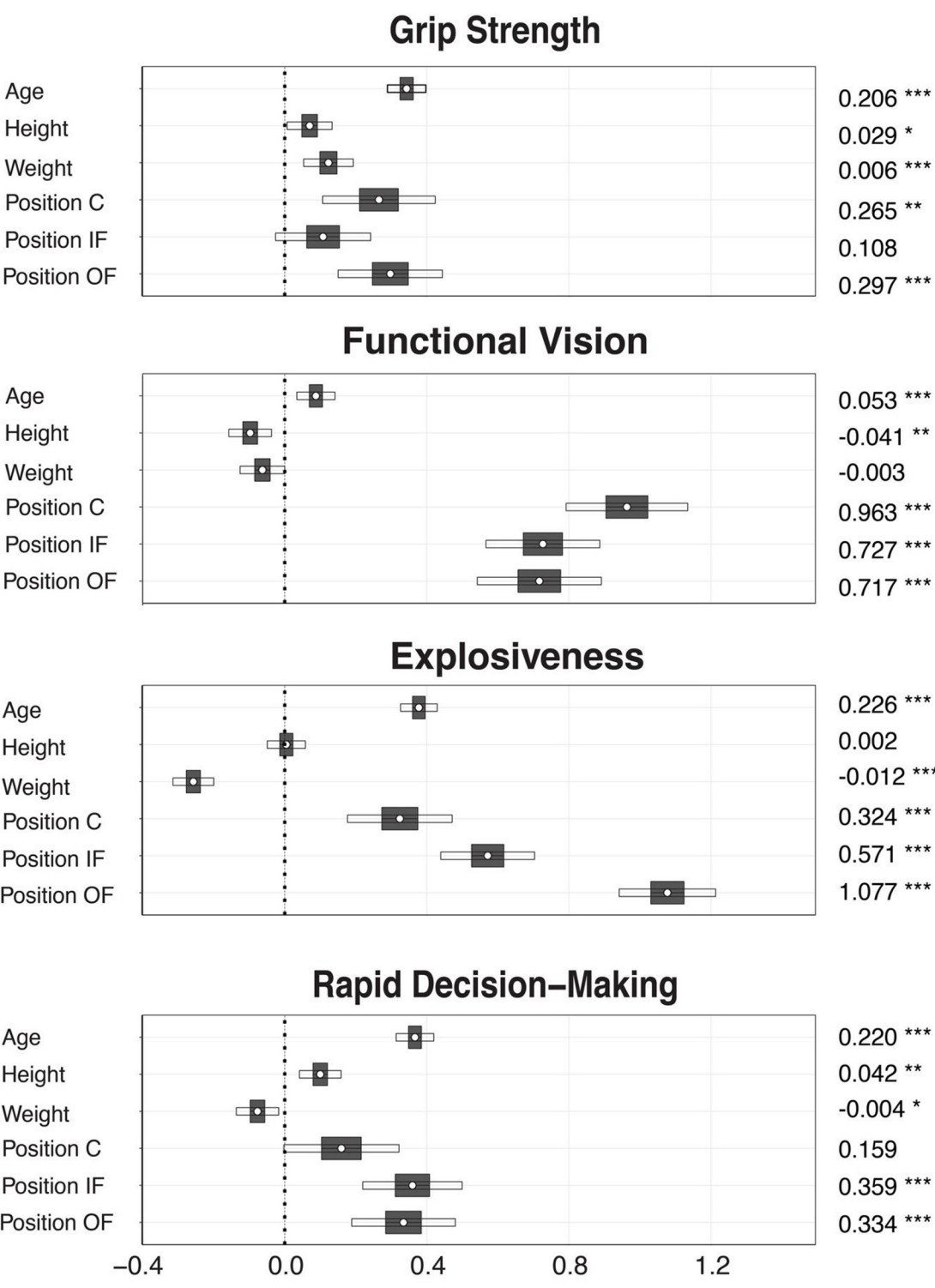

**Fig 2. Multiple regression coefficient estimates for all four models.** Mean coefficient estimates are illustrated by the central circle and listed to the right of the table along with statistical significance (*** $p < 0.001$, ** $p < 0.01$, * $p < 0.05$). Inner and outer bands represent 5% and 95% confidence intervals, respectively.

**Functional vision.** This model produced a significant ($p < 0.001$) fit with the independent variables accounting for 16% of the variance in Functional Vision factor loadings. Age was associated with an average increase in Functional Vision. Unexpectedly, height was significantly associated with an average decrease in Functional Vision. Catchers, infielders, and outfielders each had significantly better functional vision scores, compared to pitchers, with catchers showing the highest average scores.

**Explosiveness.** This model produced a significant ($p < 0.001$) fit with the independent variables accounting for 34% of the variance in Explosiveness factor loadings. The results showed that age was associated with an average increase in Explosiveness. Although height was not significant, higher weight was associated with a decrease in Explosiveness scores. Catchers, infielders, and outfielders had higher Explosiveness scores compared to pitchers, with outfielders showing the highest average scores.

**Rapid decision-making.** This model produced a significant ($p < 0.001$) fit with the independent variables accounting for 17% of the variance in Rapid Decision-Making factor loading scores. Here, age and height were associated with an average increase in Rapid Decision-Making scores, while weight was not. Catchers were not significantly different than pitchers, but infielders and outfielders had significantly higher factor scores compared to pitcher, with infielders producing the highest scores.

**Handedness and grip strength.** A two (side) by three (hand position) repeated measure analyses of variance was performed on the six hand grip strength measures with handedness as a between-subjects factor. Results indicated a significant main effect of side ($F(1, 1313) = 12.66$, $p < 0.001$, $\eta^2 = 0.01$), hand position ($F(1.8, 2366.7) = 566.3$, $p < 0.001$, $\eta^2 = 0.46$), and a side by handedness interaction ($F(1, 1313) = 6.96$, $p = 0.008$, $\eta^2 = 0.005$). There was, however, not a significant difference between-subjects for left and right handed players ($F(1,113) = 0.59$, $p = 0.44$). As illustrated in **Fig 3** and observed in post-hoc paired sample t-tests, the handedness by side interaction was driven primarily by higher grip strength scores for left handed people, measured with their left (103.86) hand versus their right hand (101.03; $t(194) = 3.56$, $p < 0.001$). Surprisingly, however, among right handed individuals, there was no significant difference between grip strength on the dominant right (103.32) and non-dominant left (103.75; $t(1119) = 1.18$, $p = 0.24$) sides. Across the three hand positions, grip scores were highest in the down (109.1), then up (101.82), then 90-degree (99.2) positions.

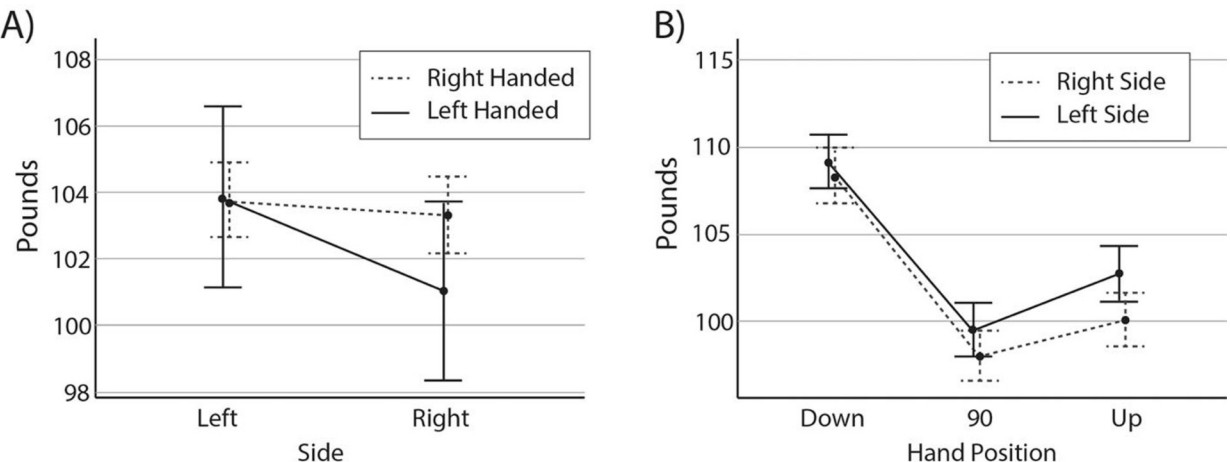

**Fig 3. Marginal means for right (dashed) and left (solid) handed athlete grip strength by side and hand position.**

## Discussion

Baseball is among the most popular sports played in America with nearly half a million athletes participating on high school teams each year according to the National Federation of State High School Associations [3]. Of these, approximately 25,000 go on to play college baseball and about 1000 are on Major League Baseball opening day rosters each year. This immense compression of talent has led to a deep desire to understand the foundational skills that underlie success and a strong investment in player development with systematic tests to measure and develop these abilities. The present study sought to contribute to this growing understanding by systematically mapping performance scores on assessments collected as part of the USA Baseball Prospect Development Pipeline to understand the structure of the underlying latent skills that were measured and to infer how position specialization manifests in the measured abilities. Through exploratory factor analysis it was observed that physical and psychomotor skills grouped into four factors that accounted for 52% of the variance. Based on the high loading from few variables these factors are interpreted as capturing latent constructs for grip strength, functional vision, explosiveness, and rapid decision-making. Multiple regression analyses performed on these factor loading scores demonstrate that these skills are significantly associated with position assignments and provide additional information above measures of age, body weight, and height.

### Psychomotor abilities

There is a growing understanding of the biomechanics of baseball specific movements and injury epistemology that has contributed to our understanding of the physical and kinematic factors that promote baseball success [27]. The present study seeks to contribute to this knowledge by evaluating movement, cognitive, and visual skills that manifest during demanding baseball drills. As such, one contribution of the present research is to help understand the latent factor structure, construed as skills critical for baseball performance, from an array of tasks that were developed for talent evaluation and selection in baseball. Although inclusion of these tasks was based on wisdoms from baseball professionals, they have rarely been scientifically examined. Through exploratory factor analyses the assessments grouped into a 4-factor model that divided clearly along conceptual dimensions, and to a lesser extent, according to the modality of measurement. Here, it was found that while hand grip strength measures, functional vision scores, and rapid decision-making each grouped into separate factors composed only of contributions from metrics measured with the same device, different kinematic tests from the 30-yard sprint, broad jump, and counter movement jump assessments grouped together to form a single factor.

### Regression discussion

The present sample consists of junior elite baseball players ages 14 to 21 years old who were invited to PDP camps. This sample of athletes is unique in that it captures an age range when young baseball players begin to specialize in sport and devote intense, year-round training in a single sport at the exclusion of other sports [28, 29]. Given that less than 1% of the young athletes in various sports achieve elite status before 17 years old [30], it is reasonable to assume that the present sample of junior baseball players would show rapid growth on many skill components in baseball but also likely to show differential growth among skill components due to the fact that baseball has demands that might mature at different times in development.

To better understand how the measured PDP assessment manifest differently among players of different height, weight, and age, as well as position specialization, multiple linear regression analyses was performed on the factors. This analysis revealed that the underlying skills

were significantly associated with position assignments providing additional information above knowledge of the players age, body weight, and height. These relationships demonstrate some key parallels with domain knowledge of baseball skills [2, 3]. Consistent with the description that infielders are quick on their feet, recognize plays, and make quick decisions; infielders here demonstrated higher "Rapid Decision-Making" scores relative to other positions. Similarly, the description that outfielders are the fastest, most powerful, and possess strong arms corresponds to the result that outfielders outperformed other positions in "Grip Strength" and "Explosiveness". In agreement with the description that catchers exhibit heightened motor-cognitive speed among other less physical factors; catchers demonstrated heightened "Functional Vision" scores relative to other positions. Further, the significant average increase in "Functional Vision" performance among catcher, infielder, and outfielder positions compared to pitcher is consistent with past observations when comparing pitchers and batters on visual skills [31].

Across all factors, the results indicated that after controlling for height, weight, and the player position, older players perform better on all tasks. This finding may relate to both expected developmental trajectories, as well as the amount of accumulated deliberate practice in baseball, which has been shown to be one of the most important determinants of sport expertise [32].

A surprising negative effect of increased height was observed on "Functional Vision" when no effect was expected. The observed negative coefficients associated with weight and height are counterintuitive as being bigger in sports like baseball is typically deemed as an advantage. An explanation for this observation is that functional vision (i.e., smooth pursuit, saccadic latency, visual speed, process speed, and eye-hand reaction time) may be an especially important "skill" for catchers, who are on average the shortest among the positions. Alternatively, for those who have a disadvantage in weight and height, superior visual and cognitive skills may compensate for the physical disadvantage compared to bigger athletes. To such a concern, functional vision might be considered a compensatory skill that makes up for the size disadvantage [33], yet it might also be a learned aspect of practice at the catcher position.

The predictive power of position variables on skills also led to some additional interesting findings. First, compared to other positions, pitchers seem to be consistently inferior at all tests, which is evidenced by the positive coefficient estimates associated with all other positions. However, given that pitchers are likely to possess unique skills in the sport, it seems to suggest an absence of some critical skill measurements in the present PDP battery. Overall, mapping the position to skills and connecting the skills to different baseball positions represent meaningful contributions to baseball, whose past research tends to focus on the critical skills associated with batting task [34, 35], representing the offensive aspect of the game and for informing the design of future sports training research. For instance, a recent pre-registered randomized placebo-controlled study focused on training baseball batters with the most promising skill elements that showed promise of transferring to field performance [36].

## Strengths and limitations

There are several strengths and limitations in this study. First, these data offer a rare and unique view into the psychomotor abilities of a very large sample of elite junior baseball players. All assessments were conducted under standardized procedures by trained PDP personnel offering high confidence in the data quality. However, assessments were collected at different event locations, opening the possibility for differences in environment that are not accounted for here. Furthermore, the eye tracking assessments and summary score aggregation across multiple sub-tasks is patent protected which poses some limitations in interpreting the

specifics of these vision measures. As an exploratory analysis of secondary data, this study is intended to generate hypotheses, which will require future studies to replicate and validate in planned analyses and controlled studies. Moreover, the exploratory factor analysis was intended to serve as a dimension reduction approach allowing the multitude of discrete measures to be condensed into a smaller number of latent abilities that could be tested in relation to player demographics and physical measures. This dimension reduction produced a small number of clear and interpretable factors, but only account for about 52% of the overall variance, leaving a considerable amount of information underexplored. Finally, it is important to note that the stratification of players into only four position groups is an oversimplification, particularly given that youth players often change position over the course of development.

## Conclusions

Ultimately, the crucial question that all stakeholders (i.e., USA Baseball, PDP, scouts, players, and coaches) wish to address is what skills aside from demographics are important in identifying elite baseball players, and what relationship these skills have with position specialization. In this study, we report there are numerous parallels between the observations of domain experts and the data driven results identified here. These findings are promising because they support the data driven approach utilized here and the usefulness of stakeholder observations from many years of domain knowledge. Future studies should validate these results, extrapolate to other sports, investigate the effects of skill-specific training methods on identified skills, and explore the relationship between assessment results and real game performance. The understanding of these metrics and future research along with PDP and other assessments may connect to numerous scientific findings and facilitate its direct application in baseball and beyond.

## Acknowledgments

The authors would like to thank USA Baseball and particularly Russell Hartford, Drew Pomeroy, and Jake Barnes who played a vital role in the collection of the data, domain expertise to guide analyses and interpretation, and support throughout the study. We are thankful to the participants in the Prospect Development Pipeline, as well as the members of the Duke University Data+ program, Malik Scott, Jack Lichtenstein, Amber Potter, Phuc Nguyen, Dr. Tim Sell, Dr. Marc Richard, Simi Bleznak, Maxwell Brown, Julia Choi, William Carson, Hunter Klein, and Joel Huber for contributing to the exploration and analysis of the larger PDP datasets provided for this research. The authors would also like to thank Liam Lochhead, and Jordan Baker for their comments on preliminary drafts of the manuscript.

## Author Contributions

**Conceptualization:** Jessica Ho, Sicong Liu, Lawrence Gregory Appelbaum.

**Formal analysis:** Jessica Ho, Sicong Liu, Lawrence Gregory Appelbaum.

**Funding acquisition:** Lawrence Gregory Appelbaum.

**Investigation:** Jessica Ho, Sicong Liu, Lawrence Gregory Appelbaum.

**Methodology:** Jessica Ho, Sicong Liu, Lawrence Gregory Appelbaum.

**Project administration:** Lawrence Gregory Appelbaum.

**Resources:** Lawrence Gregory Appelbaum.

**Supervision:** Lawrence Gregory Appelbaum.

**Validation:** Zeyu Feng.

**Visualization:** Jessica Ho.

**Writing – original draft:** Jessica Ho, Sicong Liu, Lawrence Gregory Appelbaum.

**Writing – review & editing:** Jessica Ho, Sicong Liu, Zeyu Feng, Lawrence Gregory Appelbaum.

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
