## [Decision Letter · Decision Letter 0]

8 Jun 2022

PONE-D-22-13932Psychomotor and Visual Skills Underlying Position Specialization in 1352 Elite Youth Baseball PlayersPLOS ONE

Dear Dr. Appelbaum,

Thank you for submitting your manuscript to PLOS ONE. After careful consideration, we feel that it has merit but does not fully meet PLOS ONE’s publication criteria as it currently stands. Therefore, we invite you to submit a revised version of the manuscript that addresses the points raised during the review process.

We look forward to receiving your revised manuscript.

Kind regards,

Yaodong Gu

Academic Editor

PLOS ONE

Journal Requirements:

 "This research was funded by grant support to L.G.A. through the United States Army Research Office [W911NF-15-1-0390]. " ext-link-type="uri" xlink:type="simple">https://www.arl.army.mil/who-we-are/aro/"

"All authors declare that they have no conflict of interest related to the research presented in this manuscript."

Additional Editor Comments :

Please add hypothesis in your study.

Reviewers' comments:

Reviewer's Responses to Questions

**Comments to the Author**

1. Is the manuscript technically sound, and do the data support the conclusions?

Reviewer #1: Yes

Reviewer #2: Yes

2. Has the statistical analysis been performed appropriately and rigorously? 

Reviewer #1: Yes

Reviewer #2: Yes

3. Have the authors made all data underlying the findings in their manuscript fully available?

Reviewer #1: Yes

Reviewer #2: Yes

4. Is the manuscript presented in an intelligible fashion and written in standard English?

Reviewer #1: Yes

Reviewer #2: Yes

5. Review Comments to the Author

Reviewer #1: Major issues

The manuscript’s style and format failed to meet the requirements of the journal, please review the full submission guidelines for the journal. Details are in the URL. (https://journals.plos.org/plosone/s/submission-guidelines#loc-file-format)

Minor issues

1. Manuscript should include page numbers and line numbers in the manuscript file. Use continuous line numbers (do not restart the numbering on each page).

2. In the abstract part. In the opinion of reviewer, the author provided too much background descriptions in this part, which may be too long-winded. I suggest that the authors provide more detailed descriptions of the methods, results, and conclusions of this study in this part.

3. 2.3.2. Multiple Linear Regression, please make sure the equation of was correct.

4. Table needs to be re-edited to meet journal requirements.

5. What is the author's research hypothesis, which I would suggest to be added to the last paragraph of the introduction?

6. Some small errors in the manuscript need to be corrected. For example, 4.2 Regression Discussion, “Further, the significant average increase in “Functional Vision” performance among catcher, infielder, and outfielder positions compared to pitcher is consistent with past observations when comparing pitchers and batters on visual skills (30)”, this sentence seems to be missing a period.

7. The discussion part needs more literature support, please add more references.

8. Please do check the language and grammar mistakes throughout the whole article to further improve clarity.

In summary, please make sure that your manuscript is properly prepared and formatted before submitting a revision.

Reviewer #2: This study is an interesting and valuable study, however the introduction and methods parts still need to be revised and optimized. Please ensure that the interaction factor in the repeated measures ANOVA is considered in the analysis of the results of this study. Besides, please provide more details on grouping and culling athletes.

6. PLOS authors have the option to publish the peer review history of their article (what does this mean?). If published, this will include your full peer review and any attached files.

Reviewer #1: No

Reviewer #2: **Yes: **Yuqi He

---

## [Decision Letter · Decision Letter 1]

19 Oct 2022

PONE-D-22-13932R1Psychomotor and Visual Skills Underlying Position Specialization in 1352 Elite Youth Baseball PlayersPLOS ONE

Dear Dr. Appelbaum,

Thank you for submitting your manuscript to PLOS ONE. After careful consideration, we feel that it has merit but does not fully meet PLOS ONE’s publication criteria as it currently stands. Therefore, we invite you to submit a revised version of the manuscript that addresses the points raised during the review process.

We look forward to receiving your revised manuscript.

Kind regards,

Yaodong Gu

Academic Editor

PLOS ONE

Journal Requirements:

Additional Editor Comments:

Please check some questions raised by the reviewers.

Reviewers' comments:

Reviewer's Responses to Questions

**Comments to the Author**

1. If the authors have adequately addressed your comments raised in a previous round of review and you feel that this manuscript is now acceptable for publication, you may indicate that here to bypass the “Comments to the Author” section, enter your conflict of interest statement in the “Confidential to Editor” section, and submit your "Accept" recommendation.

Reviewer #1: All comments have been addressed

Reviewer #2: (No Response)

2. Is the manuscript technically sound, and do the data support the conclusions?

Reviewer #1: Yes

Reviewer #2: Yes

3. Has the statistical analysis been performed appropriately and rigorously? 

Reviewer #1: Yes

Reviewer #2: Yes

4. Have the authors made all data underlying the findings in their manuscript fully available?

Reviewer #1: Yes

Reviewer #2: Yes

5. Is the manuscript presented in an intelligible fashion and written in standard English?

Reviewer #1: Yes

Reviewer #2: Yes

6. Review Comments to the Author

Reviewer #1: Thank you to the authors for submitting the revised manuscript. However, to enhance the quality of the manuscript, revise suggestions are given below.

1. There are some grammatical errors in the article, please pay attention to the authors to correct them carefully. For example, line 213, I think "was" should be changed to "were".

2. There are some written errors in the article, please pay attention to the authors to correct them carefully. For example, line 277, “Kolmogorov-Smirnov ps 0.01”.

3. Does Table 1 have no title? And it is recommended that the authors change the tables to the form of a three-line table.

Reviewer #2: Congratulations to the authors for the interesting work. The research is original and I enjoyed reading it. I suggest minor comments in order to improve the quality of the manuscript.

1. The structure of this manuscript is somewhat complex, and to help the reader feel more comfortable reading this study, I suggest adding a new paragraph at the end of the introduction section with a brief description of the structure of this manuscript.

2. L125-142. The description of the test process section needs to add further details, such as the requirements for athletes during the test, whether there is an interval between each test, etc. A detailed description of the test procedure is necessary for the principle of reproducibility in clinical trials.

3. L201-205. In a formal experiment, do subjects get only one chance to be tested? Will the test be allowed to be repeated if the subject makes a mistake? Please add specific details.

4. Were possible interaction issues during testing reported in the results section of this study?

5. L316-320. Please check this section for formatting issues and the header integrity of Table 1.

7. PLOS authors have the option to publish the peer review history of their article (what does this mean?). If published, this will include your full peer review and any attached files.

Reviewer #1: No

Reviewer #2: **Yes: **Yuqi He

---

## [Author Response · Author response to Decision Letter 1]

6 Nov 2022

Reviewer #1: Thank you to the authors for submitting the revised manuscript. However, to enhance the quality of the manuscript, revise suggestions are given below.

1. There are some grammatical errors in the article, please pay attention to the authors to correct them carefully. For example, line 213, I think "was" should be changed to "were".

As suggested, we have carefully read the manuscript and made several small modifications to improve the grammar. We didn’t not, however, see the instance in line 213 that the review was referencing, but did check all occurrences of the word “was” and feel that they are all correctly used in relation to singular tense concepts.

2. There are some written errors in the article, please pay attention to the authors to correct them carefully. For example, line 277, “Kolmogorov-Smirnov ps 0.01”.

We have modified the test to make it clear that we mean that “Kolmogorov-Smirnov p-values were all below 0.01”

3. Does Table 1 have no title? And it is recommended that the authors change the tables to the form of a three-line table.

Table 1 does not have a title but we adjusted the caption so that it is more clearly linked to the table. Also, we anticipate that the copy editor will change the layout of the table to fit the formatting required by PLOS One.

Reviewer #2: Congratulations to the authors for the interesting work. The research is original and I enjoyed reading it. I suggest minor comments in order to improve the quality of the manuscript.

Thank you for the positive sentiments and constructive suggestions. We have made edits to address the points raised below.

1. The structure of this manuscript is somewhat complex, and to help the reader feel more comfortable reading this study, I suggest adding a new paragraph at the end of the introduction section with a brief description of the structure of this manuscript.

Thank you for the suggestion. We have now added a paragraph at the end of the introduction to help guide the reader on the structure of the rest of the manuscript.

2. L125-142. The description of the test process section needs to add further details, such as the requirements for athletes during the test, whether there is an interval between each test, etc. A detailed description of the test procedure is necessary for the principle of reproducibility in clinical trials.

We have now added several important details about the requirements of the athletes during the task and have also included a footnote that links to the PDP instructions and calculations for these tasks.

3. L201-205. In a formal experiment, do subjects get only one chance to be tested? Will the test be allowed to be repeated if the subject makes a mistake? Please add specific details.

We have now added this information, “In the event that the athlete did not follow instructions, or the data was not accurately recorded, they were given an opportunity to repeat the assessment.”

4. Were possible interaction issues during testing reported in the results section of this study?

Given the number of tested variables in the regression models, and the presence of categorical variables (dummy variables for position), the models would have been under powered to test for all possible interactions. These were therefore not tested.

5. L316-320. Please check this section for formatting issues and the header integrity of Table 1.

We have checked the section and Table 1 and do not see any problems with formatting or the integrity of the headers. We anticipate that the journal copyeditors will format the table to abide by the journal requirements

---

## [Decision Letter · Decision Letter 2]

22 Nov 2022

Psychomotor and Visual Skills Underlying Position Specialization in 1352 Elite Youth Baseball Players

PONE-D-22-13932R2

Dear Dr. Appelbaum,

We’re pleased to inform you that your manuscript has been judged scientifically suitable for publication and will be formally accepted for publication once it meets all outstanding technical requirements.

Kind regards,

Yaodong Gu

Academic Editor

PLOS ONE

Additional Editor Comments (optional):

N/A

Reviewers' comments:

Reviewer's Responses to Questions

**Comments to the Author**

1. If the authors have adequately addressed your comments raised in a previous round of review and you feel that this manuscript is now acceptable for publication, you may indicate that here to bypass the “Comments to the Author” section, enter your conflict of interest statement in the “Confidential to Editor” section, and submit your "Accept" recommendation.

Reviewer #1: All comments have been addressed

Reviewer #2: All comments have been addressed

2. Is the manuscript technically sound, and do the data support the conclusions?

Reviewer #1: Yes

Reviewer #2: Yes

3. Has the statistical analysis been performed appropriately and rigorously? 

Reviewer #1: Yes

Reviewer #2: Yes

4. Have the authors made all data underlying the findings in their manuscript fully available?

Reviewer #1: Yes

Reviewer #2: Yes

5. Is the manuscript presented in an intelligible fashion and written in standard English?

Reviewer #1: Yes

Reviewer #2: Yes

6. Review Comments to the Author

Reviewer #1: Thank you to the authors for submitting the revised manuscript. The authors have done all the suggestions I made.

Reviewer #2: No more comments. Many thanks to the authors for answering all of the reviewers' comments, this is a valuable study.

7. PLOS authors have the option to publish the peer review history of their article (what does this mean?). If published, this will include your full peer review and any attached files.

Reviewer #1: No

Reviewer #2: **Yes: **Yuqi He

---

## [Editor Report · Acceptance letter]

6 Jan 2023

PONE-D-22-13932R2 

Psychomotor and visual skills underlying position specialization in 1352 elite youth baseball players 

Dear Dr. Appelbaum:

I'm pleased to inform you that your manuscript has been deemed suitable for publication in PLOS ONE. Congratulations! Your manuscript is now with our production department. 

Kind regards, 

on behalf of

Professor Yaodong Gu 

Academic Editor

PLOS ONE